# Effects of plasma turbulence on the nonlinear evolution of magnetic island in tokamak

Minjun J. Choi [1✉], Lāszlo Bardōczi[2], Jae-Min Kwon[1], T. S. Hahm[3], Hyeon K. Park[4], Jayhyun Kim[1], Minho Woo[1], Byoung-Ho Park[1], Gunsu S. Yun [5], Eisung Yoon[4] & George McKee[2]

Magnetic islands (MIs), resulting from a magnetic field reconnection, are ubiquitous structures in magnetized plasmas. In tokamak plasmas, recent researches suggested that the interaction between an MI and ambient turbulence can be important for the nonlinear MI evolution, but a lack of detailed experimental observations and analyses has prevented further understanding. Here, we provide comprehensive observations such as turbulence spreading into an MI and turbulence enhancement at the reconnection site, elucidating intricate effects of plasma turbulence on the nonlinear MI evolution.

[1] Korea Institute of Fusion Energy, Daejeon 34133, Republic of Korea. [2] General Atomics, P.O. Box 85608, San Diego, CA 92186-5608, USA. [3] Seoul National University, Seoul 08826, Republic of Korea. [4] Ulsan National Institute of Science and Technology, Ulsan 44919, Republic of Korea. [5] Pohang University of Science and Technology, Pohang, Gyeongbuk 37673, Republic of Korea. ✉email: mjchoi@kfe.re.kr

Magnetic island (MI) is a ubiquitous structure formed by magnetic reconnection in magnetized plasmas and associated physics has been extensively studied. In tokamak plasmas, magnetohydrodynamic instabilities[1,2] involving MIs are a serious concern since they destroy the nested structure of magnetic flux surfaces and lead to degradation of the plasma confinement. The stability of those instabilities depends on the current[1] and pressure[2] profiles of the plasma which are affected by the turbulent transport. Recent experiments[3–13] and simulations[14–24] found that an MI affects the evolution of ambient broadband fluctuations (or simply referred to as 'plasma turbulence'[25]). Plasma turbulence is significantly altered by the MI itself or a modification of the equilibrium profiles due to its presence. In brief, it increases outside the MI and decreases inside the MI following the pressure gradient. This implies that the turbulent transport around an MI can be also modified, meaning that the evolution of plasma turbulence and the evolution of the MI are coupled. However, there has been relatively little experimental research progress[7,9,11,12] in addressing the effects of plasma turbulence on the MI evolution to date.

Here, we report on experimental observations in tokamak plasmas, explaining various ways of coupling between their evolution. It is shown that the electron temperature ($T_e$) turbulence outside an MI is regulated by the flow shear and localized to the limited small region. The inhomogeneous turbulence around an MI would complicate the transport around the MI on which its stability depends. In addition, we present some observations which allow more direct intervention of plasma turbulence in the MI stability. They include turbulence spreading into the MI and the turbulence enhancement at the reconnection site, either retarding or facilitating the magnetic reconnection, respectively. These observations significantly extend our understanding of the nonlinear MI evolution in tokamak as well as provide general insights into the magnetic reconnection physics in magnetized plasmas.

## Results

**Inhomogeneous low-$k$ $T_e$ turbulence around an MI.** While turbulence around an MI has been limitedly observed in other experiments, the Korea Superconducting Tokamak Advanced Research (KSTAR)[26] experiment[8] clearly demonstrates that the increase of the low-$k$ ($k\rho_i < 1$ where $k$ is the wavenumber and $\rho_i$ is the ion Larmor radius) $T_e$ turbulence can be localized in the inner region ($r < r_s$ where $r_s$ is an MI boundary) between the X-point and O-point poloidal angles ($\theta_X < \theta < \theta_O$). In the KSTAR experiments[8], the plasma was heated by the constant ~1 MW neutral beam and an $m/n = 2/1$ MI was driven by the external $n = 1$ magnetic field perturbation for accurate measurements and analyses of small $T_e$ fluctuations around the MI using the 2D local $T_e$ diagnostics. $m$ and $n$ are the poloidal and toroidal mode numbers, respectively.

The result of 2D measurements in the inner region of the MI is summarized in Fig. 1. Figure 1a, b shows the coherence of the $T_e$ fluctuations at $\theta_a = \theta_O$ and $\theta_X < \theta_b < \theta_O$, respectively. Fluctuations whose power exceeds the significance level (red dashed line) are not observed at the O-point angle, while the strong broadband fluctuations are observed near the X-point. It means that the low-$k$ $T_e$ turbulence does not increase in the inner region at the O-point angle as the MI grows, while it increases with the local $T_e$ gradient in the inner region near the X-point angle (see below). This can be understood as an effect of the inhomogeneous flow shear around the MI. It is known that the strong flow shearing rate can suppress the turbulence growth[27,28]. Figure 1c, d shows the radial two-point measurements of the local average dispersion relation ($k_z(f)$) at $\theta_c$ and $\theta_d$, respectively. The dispersion

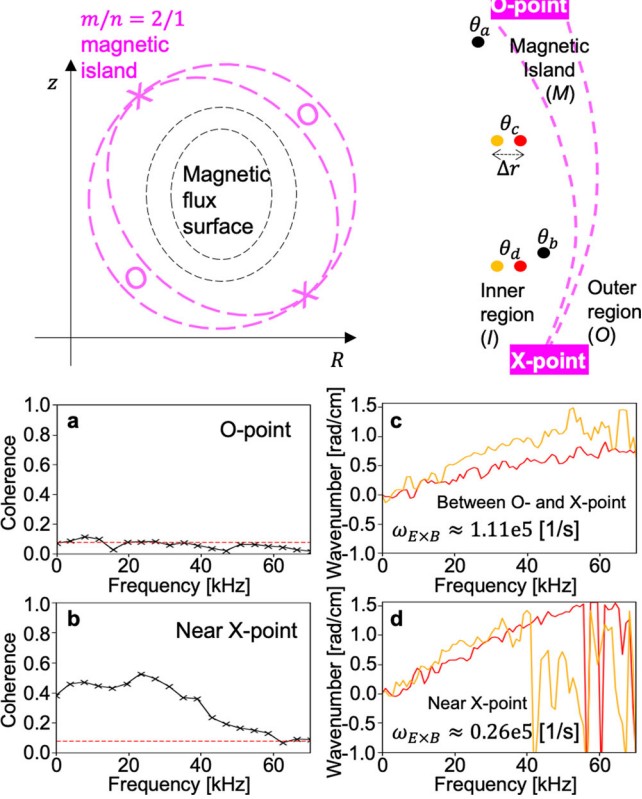

**Fig. 1 Inhomogeneous turbulence and flow around a magnetic island.** **a–d** The coherence (**a**, **b**) and local dispersion relations (**c**, **d**) of the low-$k$ $T_e$ turbulence measured at $\theta_a$, $\theta_b$, $\theta_c$, and $\theta_d$, respectively. Orange and red lines in (**c**) or (**d**) represent the two radial measurements at orange and red circles shown in the illustration, respectively.

measurement provides the laboratory frame phase velocity ($v_L = 2\pi f/k_z$) whose radial derivative can be approximated as the radial shearing rate of the $\mathbf{E} \times \mathbf{B}$ flow, i.e. $\omega_{E \times B} \approx \Delta v_{E \times B}/\Delta r \approx \Delta v_L/\Delta r$. Here, we assumed that the gradient of the turbulence intrinsic phase velocity is negligible in the small measurement region ($\Delta r \ll r$) and the flux surface squeezing effect is subdominant considering a drastic change of $\Delta v_L/\Delta r$ between $\theta_c$ and $\theta_d$. The measurements show that $\omega_{E \times B}$ is strongly increasing towards the O-point angle ($\omega_{E \times B} \approx 0.26 \pm 0.15 \times 10^5$ [1/s] at $\theta_d$ and $\omega_{E \times B} \approx 1.11 \pm 0.48 \times 10^5$ [1/s] at $\theta_c$)[8]. This explains the absence of the fluctuation in the inner region at the O-point angle since $\omega_{E \times B}$ is expected to be larger than the typical auto decorrelation rate ($10^5$ [1/s]) of tokamak plasma turbulence.

Further analysis of their temporal evolution measured in the inner region near the X-point (around $\theta_d$ in Fig. 1) reveals the dynamic nature of their interactions. In the KSTAR experiment, the $n = 1$ perturbation field was slowly increasing in time, and the field penetration and the $m/n = 2/1$ locked MI onset occurred around $t = 7.1$ sec. The normalized inverse $T_e$ gradient scale length ($a/L_{T_e}$ where $a$ is the minor radius and $L_{T_e} = |T_e/\nabla T_e|$) and the root-mean-square (RMS) amplitude of the low-$k$ normalized $T_e$ turbulence measured in time are shown in Fig. 2a. The turbulence amplitude increase is correlated with the $a/L_{T_e}$ increase around $t = 7.29$ s and its growth rate (the slope indicated by red dashed lines) becomes higher with $a/L_{T_e}$ until $t = 7.33$ s. This implies that the $T_e$ gradient is a driver of the observed turbulence[29]. However, the growth rate is reduced after $t = 7.33$ s, which cannot be explained by a considerable increase of $a/L_{T_e}$ around that time. The evolution of radial two-points

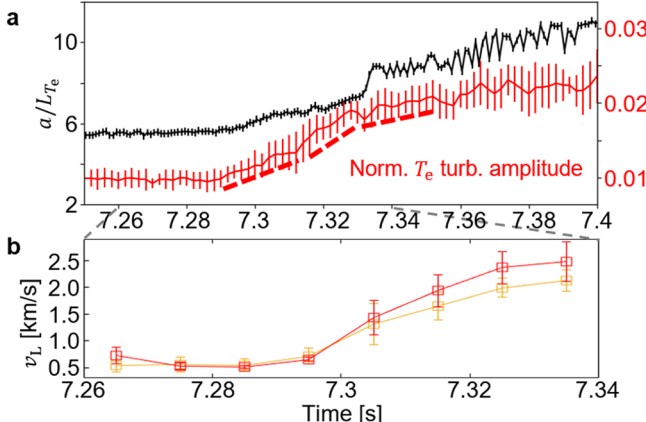

**Fig. 2 Coupled evolution of turbulence and flow. a** The normalized inverse $T_e$ gradient scale length and the root-mean-square (RMS) amplitude of the low-$k$ $T_e$ turbulence, and (**b**) the laboratory phase velocity ($v_L$) measurements at two radial positions (red and orange squares) in the inner region near the X-point. The error bar of the RMS amplitude represents the standard deviation of the measurements using different diagnostics channels. The error bar of $v_L$ represents the standard deviation of the measurements using different wavenumbers in the broadband spectrum.

measurements (orange and red squares) of $v_L$ showed that a notable flow shear (difference between red and orange squares) starts to develop around $t = 7.31$ s as shown in Fig. 2b. This suggests that the shear flow developed with the rapidly increased turbulence for $t = 7.31$–$7.33$ s in turn regulates the turbulence to a lower level than that expected from the prior trend. The nonlinear evolution of the turbulence and the flow shear around an MI would make the transport around an MI (and therefore its stability) more complicated. Their complex behavior should be carefully considered for a thorough understanding of the MI evolution. In addition, there are observations which indicate more direct effects of plasma turbulence on the MI evolution as follows.

**Turbulence spreading into an MI.** In the previous KSTAR experiment, the low-$k$ $T_e$ turbulence remains still significant near the X-point due to the increased $T_e$ gradient and the insufficient flow shear there. The increased turbulence would be localized in the inner region when its amplitude is not sufficient to overcome the strong flow shear across an MI boundary formed by the reversed flow profile from the inner to the outer region[8]. In other words, this turbulence outside an MI can spread into the MI if its amplitude is sufficiently large[30]. In the recent DIII-D[31] and HL-2A[32] experiments[12,33,34], observations which seem to result from the spreading of the density and temperature turbulence towards the O-point of an MI are reported. In addition, the nonlinear electrostatic gyrokinetic simulation based on the KSTAR plasma equilibrium and profiles showed that turbulence and the heat can spread into an MI which was initially stable region[35].

Turbulence spreading[36–38] is expected to play an important role for the evolution of an MI[39], since the accompanying heat or particle flux can change the pressure and current profile inside the MI. Detailed observation of the dynamics of turbulence spreading would be helpful to understand its effect on the MI evolution. After $t = 7.34$ s in Fig. 2a, the measurements which can be interpreted as intermittent turbulence spreading events are obtained.

Intermittent heat transport events, which are identified by sharp oscillations of $a/L_{T_e}$ in Fig. 2a, are observed when the turbulence amplitude is sufficiently large. The time traces of local

$T_e$ at the position $I$ in the inner region, at position $M$ inside the MI, and at position $O$ in the outer region are also shown in Figs. 3a,b, and c, respectively. Bold arrows in Fig. 3a, b indicate the enhanced heat transport from the inner region (where $T_e$ decreases) into the MI (where $T_e$ increases) during a single event, and the later dashed arrow in Fig. 3c indicates the rapid and global exhaust of the heat accumulated inside the MI. Four images (#1–#4) in Fig. 3d show the local relative change of the 2D $T_e$ during a single event against the quiescent period. From #1 to #3, $T_e$ increases inside the MI spontaneously. This increase can be a result of the intermittent leakage of the turbulent heat flux from the inner region through a path near the X-point where the flow shear is relatively weak. The interior magnetic topology of an MI is known to have a good perpendicular confinement character-istics[40] and the enhanced heat influx leads to the $T_e$ peaking inside the MI.

Measurements of the amplitude evolution of turbulence show that the observed transport events could result from turbulence spreading. Figure 4 shows the evolution of the low-$k$ normalized $T_e$ turbulence RMS amplitude ($|\delta T_e/\langle T_e \rangle|$) during the multiple events in high temporal resolution. Most events shown in Figs. 4a and c are found to be correlated with the turbulence amplitudes shown in Figs. 4b, d. Specifically, $|\delta T_e/\langle T_e \rangle|$ in the inner region (shown in Fig. 4b) starts to decrease with the event and $|\delta T_e/\langle T_e \rangle|$ inside the island (shown in Fig. 4d) is peaked with the event, which is consistent with a picture of turbulence spreading. The former is relatively clear in the 1st, 2nd, 4th, and 6th event and the later is in the 2nd, 3rd, 4th, 5th, and 6th event due to the finite noise contribution. This turbulence spreading behavior was also confirmed by the 2D measure-ments and it may contribute to the saturation of the local turbulence amplitude after $t = 7.35$ s in Fig. 2a. The rapid and global exhaust of the accumulated heat inside the MI, observed for #3–#4, can be attributed to the global enhancement of the heat transport with the increased turbulence inside the MI.

Although this observation is made with an externally driven MI, its evolution should also be governed by a set of equations describing the tearing instability as illustrated in reference[41]. Therefore, what is shown in Figs. 3, 4 would represent a general feature of turbulence spreading dynamics around the MI. The observation implies two beneficial effects of turbulence spreading for the MI instabilities in tokamak plasmas. Firstly, the $T_e$ peaking inside an MI can be helpful for the MI saturation through its perturbation on the Ohmic current[42] as demonstrated in the auxiliary heating experiment[43]. Secondly, for an MI driven by the neoclassical tearing mode (NTM)[44], the enhanced turbulent transport into the MI would eventually lead to the saturation at a smaller width or the stabilization[7], since it can recover the bootstrap current loss[2] when magnetic shear is positive. Indeed, the partial stabilization of an NTM with the pellet injection was observed in the recent DIII-D and KSTAR experiments[34] in which spreading of the density turbulence was regarded as a key mechanism for the stabilization.

**Turbulence increase at the X-point in the fast minor disrup-tion.** The uncontrolled MI is a serious problem in tokamak plasmas since it often results in the plasma disruption via the mode locking[45]. Here, we finally report an observation during the minor disruption of the locked MI in the KSTAR experiment, suggesting a role of the increased turbulence at the X-point in the fast plasma disruption.

Figure 5a shows the process of the fast $m/n = 2/1$ MI disruption captured by the 2D $T_e$ diagnostics. The constant external field was applied to induce a locked MI in this experiment. The normalized $T_e$ images represent the local relative

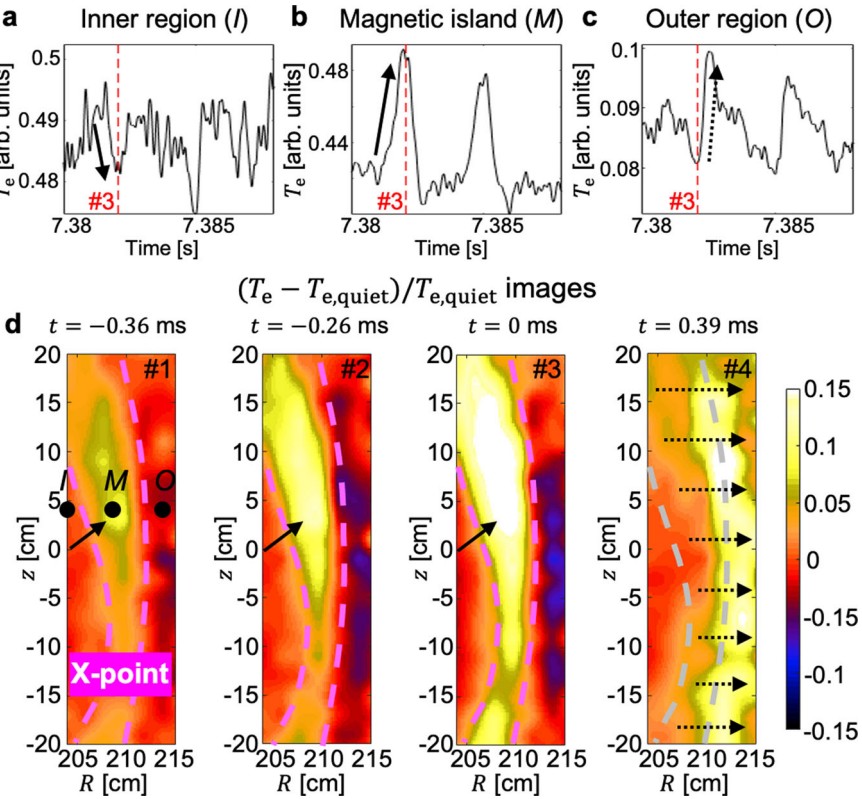

**Fig. 3 Spontaneous $T_e$ peaking inside a magnetic island. a–c,** $T_e$ measurements in time at positions $I$, $M$, and $O$ (marked in the #1 image in **d**), respectively. **d** The 2D relative change of $T_e$ around the magnetic island during a single turbulence spreading event. Absolute time of #3 is indicated by a red dashed line in (**a–c**).

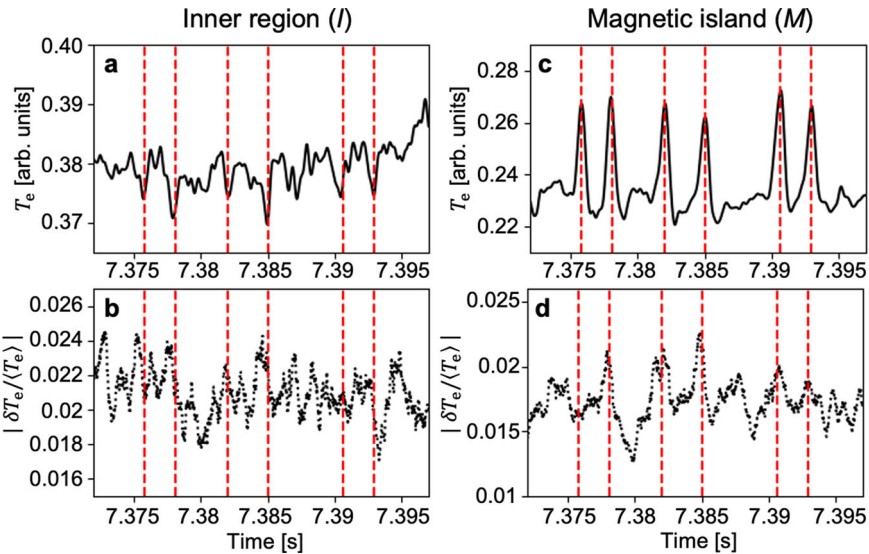

**Fig. 4 Turbulence spreading into a magnetic island. a–d** Low-pass filtered $T_e$ measurements (**a, c**) and the root-mean-square amplitudes of the normalized $T_e$ turbulence (**b, d**) in the inner region and the magnetic island region ($I$, $M$), respectively.

change of $T_e$ during the disruption against the $T_e$ at the reference time ($t = 7.673$ sec) which is just before the disruption. Purple dashed lines in the reference time image (#1) indicate the boundary of the MI identified by a prior analysis. The disruption starts with a $T_e$ collapse in the inner region as shown in the image at $+1.2$ ms (#2). The structure of this initial collapse, which is localized and poloidally symmetric across the X-point angle,

shows that the MI expands inward but the $m = 2$ topology remains by this time. The following images (#3–#4) show that $T_e$ collapses more globally, corresponding to the destruction of the MI topology via the field line stochastization in this region. Magnetic flux surfaces of the inner region are recovered around $t = 7.705$ s after a long stochastic period ($t = 7.68$–$7.705$ sec) as seen in $T_e$ measurements (black line) in Fig. 5b.

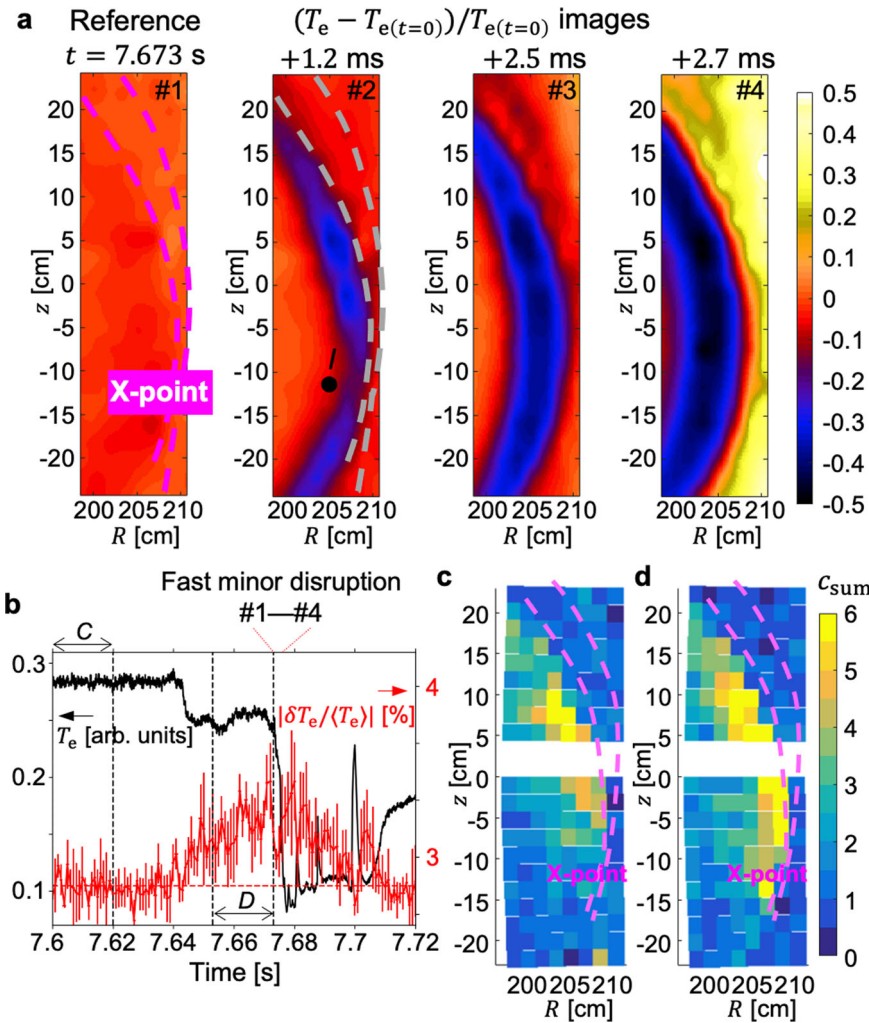

**Fig. 5 Turbulence enhancement at the reconnection site. a** The 2D relative change of $T_e$ around the magnetic island during a short period of $T_e$ collapse. The temporal range of #1–#4 are indicated by red dotted lines at the top of (**b**). **b** $T_e$ measurements (black) at position $l$ (marked on the #2 image in **a**) and the root-mean-square amplitude of the normalized $T_e$ fluctuation (red) at the X-point. **c, d** The 2D strength of the $T_e$ turbulence for the period $C$ and $D$, respectively.

$T_e$ turbulence measurements show that the correlated enhancement of turbulence at the reconnection site could contribute to the observed fast magnetic reconnection and global realignment of magnetic fields. Figures 5c, d are the images of the coherence sum of low-$k$ $T_e$ turbulence ($c_{sum} = \sum_f c(f)$ where $c(f)$ is the coherence at frequency $f$) for $C$ ($t = 7.6$–$7.62$ s) and $D$ ($t = 7.653$–$7.673$ s) of Fig. 5b, respectively. The summed coherence image is obtained by pairs of the vertically adjacent $T_e$ measurements and it represents the local strength of the $T_e$ turbulence[8]. In the period $C$, the turbulence is localized in a distant region from the X-point. However, in the period $D$, the turbulence strength increases and it expands poloidally and reaches the X-point as shown in Fig. 5d.

The temporal evolution of the RMS amplitude of the normalized $T_e$ fluctuation at the X-point is shown as a red line in Fig. 5b. The amplitude starts to increase after $t = 7.64$ s (with an unidentified small $T_e$ drop event), and it shows an increasing trend before the disruption at $t = 7.673$ s and remains significant for the stochastic period. A noise level indicated by a red dashed line could be determined as an average over the period $C$. This shows that the observed turbulence at the X-point is involved with the entire destruction process of magnetic flux surfaces and correlated with the fast reconnection event. It was found that, however, in the stochastic period the $T_e$ turbulence is no longer

localized in the small region but observed globally in the inner region, and the poloidal correlation length, which is about 4 cm before the disruption[8], becomes too small < 2 cm to be measured accurately. It is also noteworthy that this low-$k$ $T_e$ turbulence is not observed in other $m/n = 2/1$ MI-associated disruptions which have a similar level of the locked mode amplitude but occur in a longer timescale (5–10 times)[46].

## Discussion

Measurements of the low-$k$ $T_e$ turbulence and the flow shear around an MI in KSTAR experiments revealed that the turbulence is nonlinearly regulated by the flow shear development around the MI and localized to the small region near the X-point. These spatial patterns of the turbulence and the shear flow are in broad agreement with the results from many fluid and gyrokinetic simulations[15,16,19,20,22,23]. In particular, ref. [22] contains gyrokinetic simulation results based on the equilibrium and profiles of the KSTAR experiment. In spite of some limitations, a main role of the increasing flow shear toward the O-point in suppressing the turbulence was confirmed. Note that, however, different patterns of turbulence and flow were also observed in other experiments, and even a phenomenon which can be interpreted as a transition between strong and weak $\omega_{E\times B}$ states (low and high accessibility states, respectively) is observed[47].

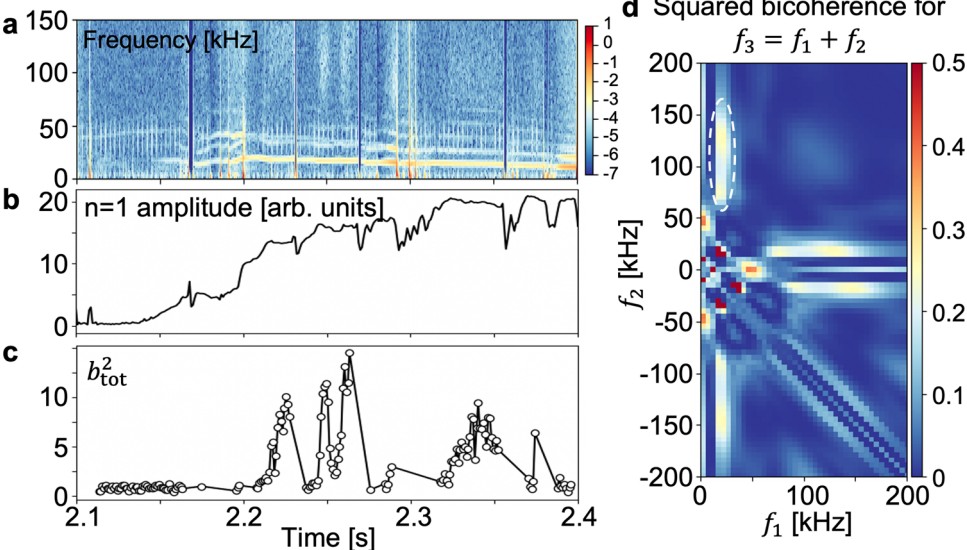

**Fig. 6 Three-wave coupling between turbulence and the neoclassical tearing mode (NTM). a** The power spectrogram of density fluctuations. **b** The $n = 1$ mode amplitude. **c** The sum of the squared bicoherence ($b_{tot}^2$) in the white dashed oval in (**d**), meaning the nonlinear coupling strength among the NTM fluctuation ($f_1$) and density turbulence ($f_2$ and $f_3$). **d** The squared bicoherence of density fluctuations for $t = 2.2475$–$2.2495$ s.

The turbulence suppression around the O-point would have a destabilizing effect on the NTM in tokamak plasmas when magnetic shear is positive. With the suppressed turbulence and reduced heat influx from the inner (hotter) region, the resulting temperature profile would be sharper across the MI boundary and flatter inside the MI than the profile without the turbulence suppression. The flatter profile inside the MI increases the bootstrap current loss and enhance the growth rate of the NTM when magnetic shear is positive. Moreover, an increasing trend of the flow shear with the island width[21,48] could form a positive feedback loop for the NTM MI growth.

When it comes to the NTM, it would be worth introducing that in the recent DIII-D experiment[34] a nonlinear three-wave coupling between the broadband (60–150 kHz) density turbulence and an $m/n = 2/1$ NTM MI is observed[11]. Figure 6a, b shows the power spectrogram of the low-$k$ ($k < 1$ cm$^{-1}$) normalized density fluctuation (measured by the beam emission spectroscopy (BES) diagnostics) and the evolution of the $n = 1$ mode amplitude (the Mirnov coil diagnostics), respectively. The squared bicoherence using the wavelet transform[49] is calculated to study the nonlinear coupling in the density fluctuations, and the result for one short period is shown in Fig. 6d. The significant squared bicoherence is observed between the ~20 kHz NTM and the 60–150 kHz turbulence, demonstrating the existence of a nonlinear three-wave coupling among them. The temporal evolution of the strength of this coupling is shown in Fig. 6c by the summed squared bicoherence ($b_{tot}^2$), which is sporadic and the strongest when the island width is intermediate not when it is the largest. Previous understanding on the interaction between density turbulence and an NTM MI relied on the pressure profile variation by the island growth[6], but this observation suggests that additional mechanism such as the nonlinear beating[50] could be also important in the NTM evolution. Further experiments and analyses would be required to better identify its role in the NTM evolution definitely.

In tokamak plasmas, the evolution of a tearing mode MI is often described by a modified Rutherford equation[51–53]. Recently, many efforts have been made to develop an advanced model which can include the multi-scale and multi-physics interaction between a large scale MI and small scale turbulence as reviewed in

reference[24]. Observations introduced in this article suggest that the model should be able to cover the nonlinear turbulence regulation by the shear flow development, the non-local effects of turbulence spreading[39], an energy exchange between a low ($m,n$) mode and high ($m,n$) modes via a three-wave coupling as well as the anomalous resistivity by turbulence.

The anomalous enhancement of resistivity by turbulence[54] has been considered as one model to explain the fast timescale of reconnection events universally observed in magnetized plasmas[55], since it can increase the dissipation rate of magnetic field and broaden the current sheet to facilitate the mass outflow. Increase of the $T_e$ turbulence at the X-point shown in Fig. 5 and its correlation with the fast plasma disruption can be a supporting evidence for that mechanism. The correlated increase of the fluctuation power during the fast magnetic reconnection in laboratory plasmas was also reported in reference[56] where the fluctuation is electromagnetic. On the other hand, another mechanism to explain the fast reconnection is based on the formation of secondary islands, or plasmoids, on a thin current sheet which becomes tearing unstable[57–60]. Successive formation of plasmoid-like structures during reconnection events were observed in various conditions such as the magnetotail[61] and laboratory plasmas[62,63]. In the KSTAR experiment[62], coalescence of plasmoid-like structures (cold bubbles) with an $m/n = 2/1$ MI lead to the explosive major disruption. The cold bubbles were formed near the X-point of an $m/n = 2/1$ MI and the MI expanded by merging with the bubbles which convect towards the plasma center[62]. It may be understandable that the low-$k$ $T_e$ turbulence discussed in this article was not observed in that case.

In summary, this article reports on various effects of plasma turbulence on the nonlinear MI evolution. Complex behavior of plasma turbulence can either retard or facilitate the magnetic reconnection process in tokamak plasmas.

## Methods

**The externally driven MI and the $T_e$ diagnostics in KSTAR.** In tokamak plasmas, MIs can be driven at the rational $q = m/n$ (safety factor where $m$ and $n$ are poloidal and toroidal mode numbers, respectively) surface by the external magnetic field perturbation which has a resonant component to that rational surface[41]. The externally driven MI is locked in the position by the external field, which allows an accurate measurement of dynamics of the MI and ambient turbulence. For the

externally driven MI experiment on KSTAR, the $n = 1$ external magnetic field perturbation was used to drive the $m/n = 2/1$ tearing mode MI at the $q = 2$ flux surface. The KSTAR plasma of the driven MI experiment has the major radius $R = 180$ cm, the minor radius $a \sim 40$ cm, the toroidal field $B_T = 2.0$–$2.2$ T, the plasma current $I_p = 600$–$700$ kA, the Spitzer resistivity $\eta \sim 1.2 \times 10^{-7}$ Ohm $\cdot$ m, and $\beta_\theta \sim 0.5$ %. Note that the constant ~1 MW neutral beam was injected to heat the plasma and it was kept in the L-mode[8].

Tokamak core plasma can be optically thick for the fundamental O-mode or the second harmonic X-mode of the electron cyclotron emission (ECE). The measured intensity of the optically thick ECE depends linearly on the local electron temperature by Kirchhoff's law and Rayleigh–Jeans law. On the other hand, the ECE frequency in tokamak plasma follows $1/R$ dependence of the toroidal field. A heterodyne detector can measure the radial profile of the electron temperature by measuring the ECE intensity selectively in frequency space. Note that in the KSTAR experiment introduced in this article the optical depth (thickness) is estimated as larger than 3 in the inner region, which enables local measurements of fine $T_e$ structures using ECE intensity measurements[64].

The inverse $T_e$ gradient scale length in this article was obtained using the $T_e$ profile measurements from the 1D ECE diagnostics. The diagnostics was calibrated via the toroidal field scan experiment as well as by comparison with the other $T_e$ profile diagnostics. Since the temporal evolution of $a/L_{T_e}$ is more important in the context of this article, the error bars of $a/L_{T_e}$ in Fig. 2a are the standard deviation of measurements in the quasi-stationary period, and they do not mean the uncertainty of absolute values.

The electron cyclotron emission imaging (ECEI) diagnostics was developed to measure the local 2D $T_e$ fluctuation in $(R, z)$ space using a vertical array of heterodyne detectors. The ECEI diagnostics on the KSTAR tokamak has 24 heterodyne detectors in vertical direction and each detector has 8 radial channels, i.e. total 192 channels[65]. It can measure the local 2D $T_e$ with a high spatial ($\Delta R \approx \Delta z \leq 2$ cm) and temporal ($\Delta t = 0.5$–$2$ $\mu$s) resolution.

This diagnostics has been utilized to study various tokamak plasma phenomena from the magnetohydrodynamic instabilities to the low-$k$ broadband turbulence[66]. For example, measurements of the local 2D $T_e$ have revealed the change of the magnetic field topology successfully in various magnetic reconnection events of magnetohydrodynamic instabilities[62,67]. On the other hand, vertically adjacent channels of the 2D diagnostics allow us to measure the power spectrum of the turbulence accurately using the cross power spectrum (or the coherence after normalization with the auto power spectra as in Figs. 1a, b) between $T_e$ measurements along the poloidal direction in which the turbulence correlation length is sufficiently long. To study the temporal evolution, the RMS amplitude of the normalized $T_e$ turbulence was measured in time using the short time cross power spectrum of vertically adjacent ECEI measurements for Fig. 2a, and the auto power spectrum for Figs. 4b,d and 5b due to its requirement of the higher resolution. For the accurate analysis, the amplitude in Fig. 2a (5b) is obtained by averaging the measurements from 12 pairs in the inner region (4 channels closest to the X-point), and error bars indicate their standard deviation. In Figs. 1, 2, measurements of phase difference between vertically adjacent ECEI channels are used to obtain the local dispersion relation ($k_z(f)$) of the turbulence and so their laboratory phase velocities ($v_L$). Error bars of $v_L$ measurements represent the standard deviation of measurements for many $k_z$s in the broadband frequency range.

**The NTM and the $n_e$ fluctuation diagnostics in DIII-D**. The $m/n = 2/1$ NTM in this article was observed in the DIII-D stationary hybrid H-mode plasma[34]. The plasma has the major radius $R = 173$ cm, the minor radius $a \sim 60$ cm, the toroidal field $B_T = 1.86$ T, the plasma current $I_p = 1400$ kA, and the normalized beta $\beta_N \sim 1.39$%.

The BES diagnostics[68] was used to measure the normalized local density fluctuation around the NTM MI. It can measure the low-$k$ ($k < 1$ cm$^{-1}$) density fluctuation with a high spatial ($\Delta R \approx \Delta z \leq 1$ cm) and temporal ($\Delta t = 1$ $\mu$s) resolution. Since the BES measurements are affected by the beam modulation or the crash event of the edge localized mode, their analyses shown in Fig. 6 are limited to periods without them.

**On the estimation of the E × B shearing rate**. The $\mathbf{E} \times \mathbf{B}$ shearing rate in general toroidal geometry[28] assuming isotropic eddy shape[69] is given by

$$
\begin{aligned}
\omega_{\mathbf{E} \times \mathbf{B}} &= \frac{(RB_\theta)^2}{B} \left| \frac{\partial}{\partial \psi} \left( \frac{E_r}{RB_\theta} \right) \right| \\
&= \frac{(RB_\theta)^2}{B} \left| \frac{\partial}{\partial \psi} \left( \frac{Bv_{\mathbf{E} \times \mathbf{B}}}{RB_\theta} \right) \right|.
\end{aligned}
\tag{1}
$$

Even in the presence of a large MI, $d\psi \approx d\psi_0 = RB_\theta dr$ is a good approximation for poloidal flux function $\psi$ at the points of flow measurements, noting that the distorted flux contours in figures are drawn after subtracting the local $\mathbf{B}_\theta$ at the rational surface where the MI exists. The flux squeezing effect due to an MI is negligible because $B_\theta \gg \delta B_\theta$, although it may seem considerable from the cartoon. Then, we have

$$
\omega_{\mathbf{E} \times \mathbf{B}} = \frac{(RB_\theta)^2}{B} \left| \frac{\partial}{\partial \psi} \left( \frac{Bv_{\mathbf{E} \times \mathbf{B}}}{RB_\theta} \right) \right| \approx \left| \frac{\partial v_{\mathbf{E} \times \mathbf{B}}}{\partial r} \right|
\tag{2}
$$

if the radial variation of $v_{\mathbf{E} \times \mathbf{B}}$ dominates over that of a geometrical factor $\frac{RB_\theta}{B}$, i.e. $\frac{1}{v_{\mathbf{E} \times \mathbf{B}}} \left| \frac{\partial v_{\mathbf{E} \times \mathbf{B}}}{\partial r} \right| \gg \left| \frac{RB_\theta}{B} \frac{\partial}{\partial r} \left( \frac{B}{RB_\theta} \right) \right|$. In general, flux squeezing effects either due to an MI or equilibrium variation of $\frac{RB_\theta}{B}$ (or $\frac{(RB_\theta)^2}{B}$) exist. But in our case, the variation of $\left| \frac{\partial v_{\mathbf{E} \times \mathbf{B}}}{\partial r} \right|$, estimated by $\left| \frac{\partial v_L}{\partial r} \right|$, is so drastic (~400 % change) between the locations $\theta_c$ and $\theta_d$ that using the expression in Eq. (2) for our estimation of $\omega_{\mathbf{E} \times \mathbf{B}}$ is justified.

**Analysis method of the nonlinear three-wave coupling**. The nonlinear three-wave coupling can be identified using the fact that the coupled waves have a well-defined phase relation[70]. The squared bicoherence used in this article to measure the degree of the three-wave coupling among $f_1$, $f_2$, and $f_3 = f_1 + f_2$ is defined as the fraction of the power at the frequency $f_3$ due to the coupling against the total power at $f_3$

$$
b^2(f_1, f_2) = \frac{|\langle X_1 X_2 X_3^* \rangle|^2}{\langle |X_1 X_2|^2 \rangle \langle |X_3|^2 \rangle},
\tag{3}
$$

where $X_1$, $X_2$, and $X_3$ are the Morlet wavelet transformation[71] coefficients of our time-series data at the corresponding frequencies and $\langle \rangle$ means a time integration[49].

## Data availability
Raw data were generated at the KSTAR facility. Derived data are available from the corresponding author upon request.

## Code availability
The data analysis codes used for the figures of this article are available via the GitHub repository https://github.com/minjunJchoi/fluctana[72,73].

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

## Acknowledgements

The authors would appreciate all the supports from the KSTAR team and the DIII-D team. One of the authors (M.J.C.) acknowledges helpful discussion with Dr. K. Ida and Dr. T. Rhodes. This research was supported by Korean Ministry of Science and ICT under NFRI R&D programs (NFRI-EN2001-11 and NFRI-EN2041-6), by National Research Foundation of Korea under NRF-2019M1A7A1A03088462, and also by the U.S. Department of Energy, Office of Science, Office of Fusion Energy Sciences, using the DIII-D National Fusion Facility, a DOE Office of Science user facility, under Award(s) DE-FC02-04ER54698. Disclaimer: This report was prepared as an account of work sponsored by an agency of the United States Government. Neither the United States Government nor any agency thereof, nor any of their employees, makes any warranty, express or implied, or assumes any legal liability or responsibility for the accuracy, completeness, or usefulness of any information, apparatus, product, or process disclosed, or represents that its use would not infringe privately owned rights. Reference herein to any specific commercial product, process, or service by trade name, trademark, manu-facturer, or otherwise does not necessarily constitute or imply its endorsement, recom-mendation, or favoring by the United States Government or any agency thereof. The views and opinions of authors expressed herein do not necessarily state or reflect those of the United States Government or any agency thereof.

## Author contributions

M.J.C. developed the central idea of the research based on discussion with L.B., J.-M.K., T.S.H., H.K.P., and E.Y. about the previous experiment conducted by J.K. in the KSTAR and L.B. in the DIII-D. M.J.C. gathered and analyzed the data and carried out the further experiment with general guidance from J.K., M.W., and B.-H.P. L.B., J.-M.K., T.S.H., H.K.P., and E.Y. contributed to the interpretation of the experimental results and the revision of the draft. G.S.Y. and G.M. contributed to the acquisition of the diagnostics data presented in this article.

## Competing interests

The authors declare no competing interests.
