## [Peer Review File · Nature Communications]

Reviewer #4 (Remarks to the Author):

The manuscript discusses a very interesting and fundamentally relevant topic, connecting the area of magnetic reconnection to plasma turbulence. In fact, the range of these results could go beyond magnetic confinement fusion. An examples are reconnection events in general, the time scales of which cannot be explained without imposing turbulence effects (however, it also has to be pointed out, that magnetic confinement devices are very special in that the strong lead magnetic field essentially reduces both reconnection and turbulence to 2-dimensional problems).

Now to the issues I see with the manuscript. Basis of the analysis is the imaging ECE diagnostic, which is indeed a very commendable piece of hardware providing high spatial and temporal resolution and covering a significant fraction of the poloidal cross-section of the KSTAR plasma. The results and the interpretation essential rely on this measurement of 2-D the electron temperature evolution. The basic assumption behind the analysis is that the spatial distribution of the electron temperature provides information about the magnetic field topology and the transport properties of the plasma. Using ECE in such a way also assumes that the plasma is optically thick, which is unquestionable here, but should be more clearly explained in the methods section. Actually, the presented correlation analysis (covering very small radial scales) already indicates that the plasma is optically thick in the relevant wavelength range. The core of the manuscript postulates a feedback mechanism derived from the experimental observations. While the explanations seem plausible, the authors actually admit themselves that their interpretation could be one of many. This in itself should not prevent publication. However, a clear distinction between arguments, which directly derive from experimental observations, and more speculative extrapolations should be made. In particular, I do not understand why the authors insist on blending their observations with NTM theory. I suspect that the NTM story was included (NTMs are used as a motivation and are part of the feedback loop), hoping to enhance the relevance of the results. I would say that the observations and the interpretation indicate an important physics mechanism which does not need the NTM story. NTMs can be mentioned (e.g. in the conclusions) as a more speculative application. The main part of the manuscript would benefit, if the main line of arguments was strengthened, focusing on the observations and what can be inferred from them directly. In this respect, I fully agree with the corresponding comments of the previous referees.

In a similar way, the discussion of the relation between island dynamic and disruptions stands out as quite speculative. While the presented IECE observations are indeed impressive, the interpretation is full of "probably due to" or "indicates". This does not fulfil the standards of a proper analysis. My recommendation is to leave this part completely out.

Overall, observations and interpretations presented in the manuscript are worth publishing in Nature Communications. However, a major revision is necessary. The main line of arguments needs to focus more clearly on the observations and what can be derived from them directly. In addition, I have a few more specific comments which should be considered before publication.

Introduction: In the general, the statement about the relation between BS-current and island growth is not correct. The loss of BS-current only acts in a positive feedback loop, leading to an increase of the magnetic island, if the magnetic shear is positive. For negative magnetic shear, the NTM mechanism does not work.

A central aspect of the results is the derivation of the flow velocity around the island. Of particular importance is the observation of the temporal and spatial evolution of the phase velocity. In this context, the authors do not discuss the effect of the magnetic field topology near magnetic islands. I wonder whether the compression of the magnetic flux, when going from the X-point of the island to the O-point, influences the result. As an approximation of the radial shearing rate two point measurements of $\Delta v/\Delta r$ are used. Figure 1 indicates that Δr is constant in real space, however, in flux coordinates this would mean that Δr changes between observation positions c and d. Could the authors please comment and explain how they deal with this issue.

When discussing transport changes manifested in the change of the electron temperature gradients around the magnetic island (e.g. figure 2b), it is important to point out that these are steady-state profiles at constant heating power (which I assume they are). Otherwise, steeper gradients could also be a result of the plasma response to an abrupt increase of the heating power (heat pulse) or a fast transport change at constant heating power, which is not located at the magnetic island. Both effects, could lead to a change of the gradients without requiring a transport change around the island. Basically, these arguments boil down to the question, whether the authors have verified, using a power balance analysis, that the steepening of the Te-gradients can be really attributed to a reduction of the electron heat transport near the magnetic island.

Another point concerning figure 2b: While the Te-profiles look smooth and the change of profile shapes seem convincing, I have a problem with the error bars. Taking them seriously, one could argue that the difference between the two profiles with magnetic island are not significant. Since a significant part of the arguments in the manuscript actually rely on the change of slope between only two points (at $\approx 1.435\text{m}$ and $\approx 1.415\text{m}$), the question of error margins must be addressed more thoroughly (possibly in the methods section).

Finally, I would like to touch a point, which to some extent was already brought up by the previous referees. The underlying experiment uses resonant magnetic field perturbations which are fixed in space. According to reference 10 the plasma is produced and maintained by low power neutral beam injection (1 MW) keeping the plasma below the H-mode threshold. Since the KSTAR NBI is quite tangential, one would expect a significant momentum input. This is, in fact, shown in figure 1 of reference 10 where the toroidal rotation velocity abruptly breaks at $\approx 7.1\text{s}$ while the $n=1$ field strength is continuously increasing. This indicates that error field screening, penetration and possibly also amplification needs to be taken into account when discussing the island dynamics. In figure 2a of the manuscript the evolution of the phase velocity is compared with increase of the error field amplitude (represented by the coils current). However, without understanding the error field penetration this can be misleading, as the amplitude of the vacuum error field may not represent the actual error field in the plasma. My question would be whether the authors have a clear understanding of the error field penetration (which also should be explained in the manuscript). Admittedly, the time at which the increase of the phase velocity sets in close to 7.3s is 200 ms after the toroidal rotation breaks, indicating the error should have penetrated sufficiently. Also, with $< 1\text{keV}$ the electron temperature is not very high. However, on the other hand, the applied field is still increasing and also the island (as represented by the Te-profiles) is evolving. It seems to me that more evidence needs to be presented to rule out that error field penetration or amplification effects are (at least partially) responsible for the observations.

Reviewer #5 (Remarks to the Author):

In the paper entitled "Effects of plasma turbulence on the nonlinear evolution of magnetic island in (a) tokamak" the authors describe a series of careful, clever experiments wherein a non-rotating magnetic island is created using static 3D error fields. A novel 2D electron-cyclotron emission diagnostic is then used to measure the electron-temperature-fluctuation characteristics in and around the induced magnetic structure. Simultaneously they have measured the local plasma parameters, including the plasma flow, in and around the island and used this information to conclude feature of the interaction between plasma turbulence and magnetic islands. These are very novel experiments. The use of a static island is a clever trick that enables high-resolution measurements with the 2D-ECE which would, no doubt, have difficulty drawing conclusions if the island were rotating.

The authors then go on to describe the mechanisms that they think are important for plasma disruptions caused by magnetic islands.

It is at this point that the authors have made what I think are unsupported conclusions. In particular they draw conclusions about the role of flow shear on the growth of a type of island called a neoclassical tearing mode. The neoclassical tearing mode is very important for tokamaks because it sets an upper-limit on the achievable pressure (and hence the fusion power) in a tokamak reactor.

Unfortunately, many neoclassical tearing modes begin their life as rotating modes. They only become static when they are very large (the precise size at which they become stationary depending on the electrical resistivity of the first wall, the intrinsic error fields in the device and the initial plasma rotation). The imposition of a static error fields – required to make the measurement – fundamentally changes the flow profile around the island. This means that conclusion regarding the effects of the flow profile cannot be readily extrapolated to the case of an initially rotating mode. This complaint was also raised by one of the preceding referees.

It is unfortunate that such a clever experiment, which measured very nicely several new features of plasma behavior in the presence of magnetic islands, does not answer this important question, but in my view the conclusions are not supported by the data. Also unfortunately, without the conclusions regarding the behavior of neoclassical tearing modes, this paper lacks sufficient impact to justify its publication in Nature Communications.

Dear Reviewers of Nature Communications,

We appreciate reviewers for providing helpful comments to improve our manuscript. We have read the comments carefully and amended our manuscript according to the suggestions. We removed the NTM feedback loop from the main results and only included arguments which are directly derived from the measurement in the main results as reviewer #4 suggested. Some discussion beyond the direct measurement is moved to the new discussion section. In response to reviewer #5's comment, we included a new result which demonstrates the importance of the turbulence in the NTM evolution more clearly (figure 5 and the corresponding text in the revision). Also, we added more explanations on discharges, diagnostics and analysis techniques in the methods section. Below are our point-by-point replies printed in black to the reviewers' comments written in blue. The corresponding changes in the revision are indicated in red.

Reviewer #4

The manuscript discusses a very interesting und fundamentally relevant topic, connecting the area of magnetic reconnection to plasma turbulence. In fact, the range of these results could go beyond magnetic confinement fusion. An examples are reconnection events in general, the time scales of which cannot be explained without imposing turbulence effects (however, it also has to be pointed out, that magnetic confinement devices are very special in that the strong lead magnetic field essentially reduces both reconnection and turbulence to 2-dimensional problems).

Now to the issues I see with the manuscript. Basis of the analysis is the imaging ECE diagnostic, which is indeed a very commendable piece of hardware providing high spatial and temporal resolution and covering a significant fraction of the poloidal cross-section of the KSTAR plasma. The results and the interpretation essential rely on this measurement of 2-D the electron temperature evolution. The basic assumption behind the analysis is that the spatial distribution of the electron temperature provides information about the magnetic field topology and the transport properties of the plasma. Using ECE in such a way also assumes that the plasma is optically thick, which is unquestionable here, but should be more clearly explained in the methods section. Actually, the presented correlation analysis (covering very small radial scales) already indicates that the plasma is optically thick in the relevant wavelength range.

> Thank you for the helpful suggestion.

+ [page 6, right column] The optical depth of the analyzed KSTAR plasma is explained in the methods section.

The core of the manuscript postulates a feedback mechanism derived from the experimental observations. While the explanations seem plausible, the authors actually admit themselves that their interpretation could be one of many. This in itself should not prevent publication. However, a clear distinction between arguments, which directly derive from experimental observations, and more speculative extrapolations should be made. In particular, I do not understand why the authors insist on blending their observations with NTM theory. I suspect that the NTM story was included (NTMs are used as a motivation and are part of the feedback loop), hoping to enhance the relevance of the results. I would say that the observations and the interpretation indicate an important physics mechanism which does not need the NTM story. NTMs can be mentioned (e.g. in the conclusions) as a more speculative application. The main part of the manuscript would benefit, if the main line of arguments was strengthened, focusing on the observations and what can be inferred from them directly. In this respect, I fully agree with the corresponding comments of the previous referees.

> Following the reviewer's suggestion, we removed the NTM feedback loop from the main result but mentioned briefly in the discussion section.

+ Figure 3 in the previous version of manuscript is removed.

+ [page 5, right column] Discussion on the NTM feedback loop is provided.

> Instead, we improved measurements of the temporal evolution of the turbulence amplitude to strengthen our argument on the flow shear regulation of turbulence around the MI.

+ [page 2, right column—page 3, left column] Measurements of the inverse Te gradient scale length and the Te turbulence amplitude (which were shown in figure 4a in the previous manuscript) are improved and added as figure 2. A paragraph describing and interpreting those measurements replaces paragraphs about the NTM feedback loop.

In a similar way, the discussion of the relation between island dynamic and disruptions stands out as quite speculative. While the presented IECE observations are indeed impressive, the interpretation is full of “probably due to” or “indicates”. This does not fulfil the standards of a proper analysis. My recommendation is to leave this part completely out.

> We improved figure 6 and removed some speculative expressions, and focused on what we can see directly from the measurement. We emphasized the correlation between the Te turbulence and the fast disruption event by providing more measurements.

+ [page 4, right column] Sentences which explain the disruption process captured by the 2D Te measurements are improved.

+ [page 4, right column—page 6, right column and figure 6b] More information on the turbulence measurement and analysis, including the high temporal resolution measurement of the Te turbulence, is provided to emphasize the correlation between the increased turbulence at the X-point and the disruption process involving the fast reconnection and the field stochasticization.

> Regarding to referee #4’s comments in the beginning, i.e. “*An examples are reconnection events in general, the time scales of which cannot be explained without imposing turbulence effects*”, we think that this observation can be another evidence that the turbulence is closely in the fast reconnection events.

+ [page 5, right column—page 6, left column] To link our observation to the previously reported similar observations for general understanding, a paragraph of more discussion is added.

Overall, observations and interpretations presented in the manuscript are worth publishing in Nature Communications. However, a major revision is necessary. The main line of arguments needs to focus more clearly on the observations and what can be derived from them directly. In addition, I have a few more specific comments which should be considered before publication.

Introduction: In the general, the statement about the relation between BS-current and island growth is not correct. The loss of BS-current only acts in a positive feedback loop, leading to an increase of the magnetic island, if the magnetic shear is positive. For negative magnetic shear, the NTM mechanism does not work.

> We rewrote the introduction and the NTM part was removed from the main results.

+ [page 1, left and right columns] The first and second paragraph are rewritten according to the changes.

A central aspect of the results is the derivation of the flow velocity around the island. Of particular importance is the observation of the temporal and spatial evolution of the phase velocity. In this context, the authors do not discuss the effect of the magnetic field topology near magnetic islands. I wonder whether the compression of the magnetic flux, when going from the X-point of the island to the O-point, influences the result. As an approximation of the radial shearing rate two point measurements of $\Delta v/\Delta r$ are used. Figure 1 indicates that Δr is constant in real space, however, in flux coordinates this would mean that Δr changes between observation positions c and d. Could the authors please comment and explain how they deal with this issue.

> The $E \times B$ shearing rate in general toroidal geometry [1] assuming isotropic eddy shape [2] is given by

$$\omega_{E \times B} = \frac{(RB_\theta)^2}{B} \left| \frac{\partial}{\partial \psi} \left(\frac{E_r}{RB_\theta} \right) \right| = \frac{(RB_\theta)^2}{B} \left| \frac{\partial}{\partial \psi} \left(\frac{Bv_{E \times B}}{RB_\theta} \right) \right| \quad (1)$$

Even in the presence of a large MI, $d\psi \approx d\psi_0 = RB_\theta dr$ is a good approximation for poloidal flux function ψ , noting that the distorted flux contours in figures are drawn after subtracting the local \vec{B}_θ at the rational surface where MI exists. The flux squeezing effect due to an MI is negligible because $B_\theta \gg \delta B_\theta$, although it may seem considerable from the cartoon. Then, we have

$$\omega_{E \times B} = \frac{RB\theta}{B} \left| \frac{\partial}{\partial r} \left(\frac{Bv_{E \times B}}{RB\theta} \right) \right| \approx \left| \frac{\partial v_{E \times B}}{\partial r} \right| \quad (2)$$

if the radial variation of $v_{E \times B}$ dominates over that of a geometrical factor $\frac{RB\theta}{B}$, i.e. $\frac{1}{v_{E \times B}} \left| \frac{\partial v_{E \times B}}{\partial r} \right| \gg \left| \frac{RB\theta}{B} \frac{\partial}{\partial r} \left(\frac{B}{RB\theta} \right) \right|$.

In general, flux squeezing effects either due to an MI or equilibrium variation of $\frac{RB\theta}{B}$ (or $\frac{(RB\theta)^2}{B}$) exist. But in our case, the variation of $\left| \frac{\partial v_{E \times B}}{\partial r} \right|$, estimated by $\left| \frac{\partial v_L}{\partial r} \right|$, is so drastic ($\sim 400\%$ change) between the locations θ_c and θ_d that using the expression in Eq. (2) for our estimation of $\omega_{E \times B}$ is justified. Note that Eq. (1) has been derived for the case where Φ_0 is a flux function, i.e. $\Phi_0 = \Phi_0(\psi)$. If that assumption is valid, the flux squeezing varies according to $\omega_{E \times B} = \frac{(RB\theta)^2}{B} \left| \frac{\partial^2}{\partial \psi^2} \Phi_0 \right| \propto \frac{(RB\theta)^2}{B}$. But in our case of experiment, this approximation is not strictly valid for a complicated $E \times B$ flow pattern around an MI. It's worth noting that, from different physical motivations, the shearing criterion for non-flux function $\Phi_0 = \Phi_0(\psi, \theta)$ has been considered [3]. In that case, different components of flow shear can also contribute to the turbulence suppression condition. Nonetheless, for the cases we analyzed in this work, v_L is predominantly in the poloidal direction with its sharp radial variation and any further refinement of the shearing rate beyond Eq. (2) seems unnecessary for illustrating our main points.

[1] T. S. Hahm and K. H. Burrell, PoP 2, 1648 (1995)

[2] K. H. Burrell, PoP 4, 1499 (1997)

[3] T. S. Hahm and K. H. Burrell, PPCF 38, 1427 (1996)

+ [page 2, right column and page 7, right column] We added some sentences to explain our assumption used in the shearing rate estimation in main text. Also, the first paragraph of the above reply was included as one subsection in the methods section.

> In fact, to avoid any unwanted geometrical effects, we selected θ_c and θ_d as $z = 5$ cm and $z = -5$ cm at the same R , respectively. We could also compare the degree of squeezing of Te contours at two locations, and the difference was too small to be measured clearly.

When discussing transport changes manifested in the change of the electron temperature gradients around the magnetic island (e.g. figure 2b), it is important to point out that these are steady-state profiles at constant heating power (which I assume they are). Otherwise, steeper gradients could also be a result of the plasma response to an abrupt increase of the heating power (heat pulse) or a fast transport change at constant heating power, which is not located at the magnetic island. Both effects, could lead to a change of the gradients without requiring a transport change around the island. Basically, these arguments boil down to the question, whether the authors have verified, using a power balance analysis, that the steepening of the Te-gradients can be really attributed to a reduction of the electron heat transport near the magnetic island.

> Thank you for helpful suggestion.

+ [page 2, left column and page 6, right column] We state explicitly in the main text as well as in the methods section that the constant ~ 1 MW neutral beam was injected in the KSTAR experiment.

> The power balance analysis would require a modeling of the island transport which is found to be non-trivial. Instead, as mentioned before, we focused on the nonlinear behavior of the Te turbulence and the shear flow based on the direct measurement and discussed its implication on the MI stability.

Another point concerning figure 2b: While the Te-profiles look smooth and the change of profile shapes seem convincing, I have a problem with the error bars. Taking them seriously, one could argue that the difference between the two profiles with magnetic island are not significant. Since a significant part of the arguments in the manuscript actually rely on the change of slope between only two points (at ≈ 1.435 m and ≈ 1.415 m), the question of error margins must be addressed more thoroughly (possibly in the methods section).

> We're sorry for the confusion. Please note that error bars of the Te profile measurements do not represent real (calibration) errors. It was not easy to estimate error bars of Te profile measurements, since those profiles (whose raw data given by 1D ECE diagnostics) are obtained by result of combined efforts of absolute and cross calibration. Here, the cross calibration means both the cross-channel (using

plasma itself) and cross-diagnostics (using the Thomson diagnostics) calibration. In figure 2 of the previous manuscript, we just added error bars to indicate 10% range of absolute Te values, but we believe that real errors would be smaller considering the flatness of the Te profile inside the island.

> Calibration errors can affect the absolute value of the Te gradient, but what it matters in our context is a relative (temporal) change of the Te gradient which is not affected by the calibration error.

> In the revision, we removed the Te profile measurements since we want to focus on the dynamics of the turbulence and flow shear. The measurement of the inverse Te gradient scale length is added in figure 2a and explanation of its error is provided in the methods section.

+ [page 7, left column] We added explanation for error bars of the inverse Te gradient scale length. We also provided brief explanations of data analysis process for figures included in this article.

Finally, I would like to touch a point, which to some extent was already brought up by the previous referees. The underlying experiment uses resonant magnetic field perturbations which are fixed in space. According to reference 10 the plasma is produced and maintained by low power neutral beam injection (1 MW) keeping the plasma below the H-mode threshold. Since the KSTAR NBI is quite tangential, one would expect a significant momentum input. This is, in fact, shown in figure 1 of reference 10 where the toroidal rotation velocity abruptly breaks at ≈ 7.1 s while the $n=1$ field strength is continuously increasing. This indicates that error field screening, penetration and possibly also amplification needs to be taken into account when discussing the island dynamics. In figure 2a of the manuscript the evolution of the phase velocity is compared with increase of the error field amplitude (represented by the coils current). However, without understanding the error field penetration this can be misleading, as the amplitude of the vacuum error field may not represent the actual error field in the plasma.

> We fully agree with the reviewer's comment. We removed the vacuum field plot in figure 2 to avoid any possibly misleading information about the real island width which we cannot measure directly.

My question would be whether the authors have a clear understanding of the error field penetration (which also should be explained in the manuscript). Admittedly, the time at which the increase of the phase velocity sets in close to 7.3s is 200 ms after the toroidal rotation breaks, indicating the error should have penetrated sufficiently. Also, with <1 keV the electron temperature is not very high. However, on the other hand, the applied field is still increasing and also the island (as represented by the Te-profiles) is evolving. It seems to me that more evidence needs to be presented to rule out that error field penetration or amplification effects are (at least partially) responsible for the observations.

> We agree that the plasma response to the external field should be considered to understand the forced island evolution and transport around it. Unfortunately, it was not possible for us to do such a complicated analysis accurately with many unknowns. So, we removed any discussion/comment about the width of the magnetic island which cannot be measured directly, but focused on the temporal behavior of the Te gradient, the Te turbulence and the flow shear. As described in the corresponding main text, we suggested that their nonlinear behavior could be understood as the turbulence regulation by the developed flow shear.

> Although we don't have a precise local measurement of the island width, there are the magnetic diagnostics which can measure the locked mode amplitude. But, it is not a local measurement either. Nevertheless, we checked those signals, and they show the saturation or a more or less weak linear increase for the analysis period of figure 2.

Reviewer #5

In the paper entitled "Effects of plasma turbulence on the nonlinear evolution of magnetic island in (a) tokamak" the authors describe a series of careful, clever experiments wherein a non-rotating magnetic island is created using static 3D error fields. A novel 2D electron-cyclotron emission diagnostic is then used to measure the electron-temperature-fluctuation characteristics in and around the induced magnetic structure. Simultaneously they have measured the local plasma parameters, including the plasma flow, in and around the island and used this information to conclude feature of the interaction between plasma

turbulence and magnetic islands. These are very novel experiments. The use of a static island is a clever trick that enables high-resolution measurements with the 2D-ECE which would, no doubt, have difficulty drawing conclusions if the island were rotating.

The authors then go on to describe the mechanisms that they think are important for plasma disruptions caused by magnetic islands.

It is at this point that the authors have made what I think are unsupported conclusions. In particular they draw conclusions about the role of flow shear on the growth of a type of island called a neoclassical tearing mode. The neoclassical tearing mode is very important for tokamaks because it sets an upper-limit on the achievable pressure (and hence the fusion power) in a tokamak reactor.

Unfortunately, many neoclassical tearing modes begin their life as rotating modes. They only become static when they are very large (the precise size at which they become stationary depending on the electrical resistivity of the first wall, the intrinsic error fields in the device and the initial plasma rotation). The imposition of a static error fields – required to make the measurement – fundamentally changes the flow profile around the island. This means that conclusion regarding the effects of the flow profile cannot be readily extrapolated to the case of an initially rotating mode. This complaint was also raised by one of the preceding referees.

It is unfortunate that such a clever experiment, which measured very nicely several new features of plasma behavior in the presence of magnetic islands, does not answer this important question, but in my view the conclusions are not supported by the data. Also unfortunately, without the conclusions regarding the behavior of neoclassical tearing modes, this paper lacks sufficient impact to justify its publication in Nature Communications.

> In replace of the NTM feedback loop, we added new result from our recent DIII-D NTM experiment. We observed a clear three-wave coupling among the density turbulence and the NTM, which implying the nonlinear energy exchange among them and demonstrating the importance of the density turbulence in the NTM growth. We believe that this additional result can fulfill the reviewer's requirement.

+ [page 4, left column—page 4, right column and page 7, right column] Section entitled 'Nonlinear mode coupling between turbulence and the NTM' is added with figure 5. The NTM MI introduced in that section was initially rotating and the analysis is done for the initial growing phase of the rotating NTM. Please note that the analysis method (bicoherence) is described briefly in the methods section and in detail in reference 65.

> We also provided the measurement which shows the importance of the turbulence existence in the fast locked mode disruption. The NTM is critical since it can not only degrade the confinement but also lead to the disruption via the mode locking. Although the measurement was obtained in the externally driven locked mode, what we can learn from that would be applicable to the NTM-driven locked mode disruption.

Reviewer #4 (Remarks to the Author):

The manuscript was completely restructured and in parts rewritten. The first and in my opinion essential part about the turbulence dynamics around the magnetic island is largely satisfactory (first two subchapters "Inhomogeneous low-k Te turbulence around MI" and "Turbulence spreading into an MI"). Like in most of the manuscript, figures could be improved by putting consistent labels or annotations into the figures. As an example, it would ease the reading very much if for instance "o-point" and "between o-point and x-point" was directly written into figures 1a and 1b. It is also confusing that the letters a, b, c are used as indices for the angular position theta and at the same time for labelling the figure (while this coincides with figure 1a and 1b, it does not for figure 1c and 1d). Figure 2b shows the v_L traces of two radial positions. It should be indicated where they are in figure 1. By the way, similar confusion arises with other figures. Similar improvements should be made for figures 3 and 4. Besides, it is not entirely clear where position "c" is in figure 3. Coming to the chapter about "Nonlinear mode coupling between turbulence and the NTM", this chapter is been introduced as a response to the criticism of Referee #5. While the observations certainly deserve to be published, the inclusion of another observation using a different tokamak (DIII-D) and a different method (beam emission spectroscopy now measuring the density and not temperature fluctuations) looks like a makeshift attempt to maintain by all means the NTM context. The three-wave coupling is a nice result, however, this observation raises more questions than it answers.

The last results chapter about the observation of disruptions has improved. The context is now clear, however parts need more discussion or better explanations. Purely looking at figure 6, it is not clear when the disruption really takes place. It would be helpful, if the period of the thermal quench and the current quench were indicated. Actually, figure 6b shows a Te crash, however what confuses me is that Te rises again at 7.7 sec (I noticed that Te is plotted on a logarithmic scale). As a side remark, parts of figure 6 are too small, such as the insert of figure 6b. In addition, the bigger frame of figure 6b contains only three dashed lines while four time points are analysed. In the discussion of the disruption dynamics, the authors mention "stochastic" periods without explaining how they measure stochasticity. Or are these purely assumptions? This claim should be substantiated.

All in all, I am afraid to say that another revision is necessary. In particular, the new part about NTMs either needs significant expansion or should be left out.

Reviewer #5 (Remarks to the Author):

The changes made by the author have alleviated the primary concerns raised in the previous review. There no longer appears to be any major unsupported conclusions. In order to address my previous complaint, that the paper lacked impact sufficient to justify publication without the connection to neoclassical tearing modes, the authors have added a new section on the interactions of rotating modes with turbulence. This new section demonstrates the existence of an interaction by calculating the bi-coherence between the turbulence and the mode. Unfortunately the existence of a beat wave does not indicate a role of that beat wave in the growth of the mode(s). Therefore, the authors have not demonstrated that the turbulence-tearing mode interaction is important in determining neoclassical tearing mode growth (in fact, the authors don't claim this either). This is an interesting paper that deserves to be published, but does not represent a topic of broad interest outside the fusion MHD community.

Dear Reviewers of Nature Communications,

We appreciate both reviewers for providing helpful comments. In this revision, we have put efforts into the improvement of figures and associated discussion. Based on reviewer #4's comments, we have taken out the NTM part from the main results so as to focus on the Te turbulence dynamics around an MI. Now, the NTM part which is reduced and improved based on reviewer #5's comments appear in the discussion section. Below are our point-by-point replies printed in black to the reviewers' comments written in blue. The corresponding changes in the revision are described in red.

Reviewer #4

The manuscript was completely restructured and in parts rewritten. The first and in my opinion essential part about the turbulence dynamics around the magnetic island is largely satisfactory (first two subchapters "Inhomogeneous low-k Te turbulence around MI" and "Turbulence spreading into an MI").

> We would like to acknowledge many constructive comments from reviewer #4 in the first review which lead to a significant improvement.

Like in most of the manuscript, figures could be improved by putting consistent labels or annotations into the figures. As an example, it would ease the reading very much if for instance "o-point" and "between o-point and x-point" was directly written into figures 1a and 1b. It is also confusing that the letters a, b, c are used as indices for the angular position theta and at the same time for labelling the figure (while this coincides with figure 1a and 1b, it does not for figure 1c and 1d). Figure 2b shows the v_L traces of two radial positions. It should be indicated where they are in figure 1

> We thank reviewer #4 for the helpful comment.

+ [page 2, figure 1] We put annotations into the figures as suggested.

+ [page 2, figure 2; page 2, right column] We modified the caption and added some words to the main text. Please understand that the flow in figure 2b is one representative measurement near theta_d while the turbulence amplitude in figure 2a is an average of 12 measurements around theta_d. The evolution of all flow measurements shows a similar behavior as in figure 2b, but obtaining an average flow (along the poloidal direction) was difficult due to the insufficient diagnostic resolution.

By the way, similar confusion arises with other figures. Similar improvements should be made for figures 3 and 4. Besides, it is not entirely clear where position "c" is in figure 3.

> Thanks for the helpful comment. Please find that position 'c' (now 'O') is marked in the first image of figure 3d ($t=-0.36$ ms).

+ [page 3, figure 3] We changed our labels (*a*, *b*, *c*) to (*I*, *M*, *O*) which represent the inner region, the magnetic island region and the outer region, respectively. They are also used in figure 1 (illustration), figure 4 and figure 5 for consistency.

+ [page 3, figure 4] We improved the caption to avoid the confusion.

> In figure 4, we provided representative measurements obtained at two positions *I* and *M*, respectively. Please find that we had checked the change of turbulence amplitude during the turbulence spreading event in 2D as shown below. The auto-power calculation was used to capture this transient behavior and so it contains some amount of noise contribution. Nonetheless, we could tell that it overall decreases in the inner region and increases inside the magnetic island as shown in figure 4b and 4d, respectively.

Coming to the chapter about “Nonlinear mode coupling between turbulence and the NTM”, this chapter is been introduced as a response to the criticism of Referee #5. While the observations certainly deserve to be published, the inclusion of another observation using a different tokamak (DIII-D) and a different method (beam emission spectroscopy now measuring the density and not temperature fluctuations) looks like a makeshift attempt to maintain by all means the NTM context. The three-wave coupling is a nice result, however, this observation raises more questions than it answers.

> We agree that this NTM part needs more investigation, and have stated so in the discussion.

The last results chapter about the observation of disruptions has improved. The context is now clear, however parts need more discussion or better explanations. Purely looking at figure 6, it is not clear when the disruption really takes place. It would be helpful, if the period of the thermal quench and the current quench very indicated. Actually, figure 6b shows a T_e crash, however what confuses me is that T_e rises again at 7.7 sec (I noticed that T_e is plotted on a logarithmic scale). As a side remark, parts of figure 6 are too small, such as the insert of figure 6b. In addition, the bigger frame of figure 6b contains only three dashed lines while four time points are analysed.

> The disruption introduced in this article is the *minor* disruption and the plasma is recovered after some (stochastic) period. We’re sorry for the confusion. Please also find that the T_e was plotted on a linear scale in arbitrary unit.

+ [page 4, left column] We added ‘minor’ to the title of subsection.

> Please find that timings of the images (relative time against $t=7.673$ sec) are written at the top of each image.

+ [page 4, figure 5] We removed the ‘insert’ plot of figure (now) 5b and indicated a short period of T_e collapse (minor disruption captured by figure 5a) with red dotted lines and annotation.

In the discussion of the disruption dynamics, the authors mention “stochastic” periods without explaining how they measure stochasticity. Or are these purely assumptions? This claim should be substantiated.

> We reported three observations. Firstly, the magnetic island ($m=2$) topology is destructed with the fast T_e collapse as shown in the #3 image in figure 5a. Secondly, the T_e (figure 5b) remained very low for a long period after the collapse event, which implies the worse confinement for that period. Thirdly, the significant fluctuation is observed in the broader region without a coherent structure. We thought that all these observations are in accord with the appearance of the broad stochastic region.

All in all, I am afraid to say that another revision is necessary. In particular, the new part about NTMs either needs significant expansion or should be left out.

> The NTM part is taken out of the main result. However, the analysis of the nonlinear coupling of turbulence and NTM from BES measurements fits the subject of this article and will strengthen this manuscript. So, we included a reduced and improved version in the discussion section. Please also see our reply to reviewer #5’s comment regarding to this part.

+ [page 5, left and right columns] We briefly introduced the NTM results with the remark on the future work.

Reviewer #5

The changes made by the author have alleviated the primary concerns raised in the previous review. There no longer appears to be any major unsupported conclusions. In order to address my previous complaint, that the paper lacked impact sufficient to justify publication without the connection to neoclassical tearing modes, the authors have added a new section on the interactions of rotating modes with turbulence. This new section demonstrates the existence of an interaction by calculating the bicoherence between the turbulence and the mode.

> We would like to acknowledge many constructive comments from reviewer #5 in the first review which lead to a significant improvement.

Unfortunately the existence of a beat wave does not indicate a role of that beat wave in the growth of the mode(s). Therefore, the authors have not demonstrated that the turbulence-tearing mode interaction is important in determining neoclassical tearing mode growth (in fact, the authors don't claim this either).

> We agree that this NTM part needs more investigation, and have stated so in the discussion.

> In fact, we did further analyze the direction of energy transfer in the three-wave coupling based some model [Ritz, Powers and Bengtson, Physics of Fluids B 1, 153 1989 and Dudok de Wit et al., Journal of Geophysical Research 104, 17079 (1999)] as shown below.

This indicates a nonlinear energy flow from the 80—120 kHz turbulence to the ~20 kHz NTM (and to some high >130 kHz frequency components). But, it would take more effort to access the validity of assumptions used in this analysis, and we decided conservatively to present the bicoherence result only which is robust until we have gathered more data and results.

> As pointed out by reviewer #5, however, we have demonstrated the existence of three-wave coupling clearly using the bicoherence result.

> Even if the NTM were playing as a quiet medium in this three-wave coupling, the background turbulence and transport would be changed as a result of the coupling. Therefore, we think that observation of the three-wave coupling itself demonstrates the importance of interaction with turbulence in the evolution of the NTM. The NTM will gain or lose its energy directly or it will be affected by the change of background turbulence and transport.

+ [page 5, left and right columns] We included a reduced and improved version of the NTM part in the discussion with the remark on the future work, reflecting reviewer #5's comments.

This is an interesting paper that deserves to be published, but does not represent a topic of broad interest outside the fusion MHD community.

> We think that the NTM is a topic of interest mainly for the fusion community. But, the results such as the turbulence enhancement at the reconnection site during the fast reconnection and the non-locality of turbulence-MI interaction through the turbulence spreading would be a topic of high interest for the broader scientific community.

Reviewer #4 (Remarks to the Author):

Considering the changes made and the additional explanations given by the authors, I have no further comments. Regarding the relevance of the topic for a publication in Nature Communications, the topic admittedly focuses on fusion research, however, with a much broader range or context, which should be of interest for other research fields dealing with e.g. turbulence or magnetic reconnection. In conclusion, I now recommend the manuscript for publication.